# Extending Quantum Probability from Real Axis to Complex Plane

**DOI:** 10.3390/e23020210

**Published:** 2021-02-08

**Authors:** Ciann-Dong Yang, Shiang-Yi Han

**Affiliations:** 1Department of Aeronautics and Astronautics, National Cheng Kung University, Tainan 701, Taiwan; cdyang@mail.ncku.edu.tw; 2Department of Applied Physics, National University of Kaohsiung, Kaohsiung 811, Taiwan

**Keywords:** complex stochastic differential equation, complex Fokker–Planck equation, quantum trajectory, complex probability, optimal quantum guidance law

## Abstract

Probability is an important question in the ontological interpretation of quantum mechanics. It has been discussed in some trajectory interpretations such as Bohmian mechanics and stochastic mechanics. New questions arise when the probability domain extends to the complex space, including the generation of complex trajectory, the definition of the complex probability, and the relation of the complex probability to the quantum probability. The complex treatment proposed in this article applies the optimal quantum guidance law to derive the stochastic differential equation governing a particle’s random motion in the complex plane. The probability distribution $\rho_{c}\left(t,x,y\right)$ of the particle’s position over the complex plane z=x+iy is formed by an ensemble of the complex quantum random trajectories, which are solved from the complex stochastic differential equation. Meanwhile, the probability distribution $\rho_{c}\left(t,x,y\right)$ is verified by the solution of the complex Fokker–Planck equation. It is shown that quantum probability ${\left|\Psi \right|}^{2}$ and classical probability can be integrated under the framework of complex probability $\rho_{c}\left(t,x,y\right)$, such that they can both be derived from $\rho_{c}\left(t,x,y\right)$ by different statistical ways of collecting spatial points.

## 1. Introduction

Probability is the most subtle setting in quantum mechanics which extracts information from the abstract complex wave function. Quantum mechanics opened a new age of technology and led the revolution of computing with the significant invention of transistors. There is no doubt that quantum mechanics totally changed our daily life, even though we have no idea why it works in that way and why it has so many mysterious properties. We are now in a position to develop some leading technology such as quantum control, quantum computing, quantum computers, and so on. Some of the latest inventions might transcend the quantum barrier and approach the limit of the classical boundary, such that more fundamental knowledge of the microscopic world might be required. 

For more than ten decades, scientists have attempted to find the relationship between quantum mechanics and classical mechanics. Hidden-variable theories introduce unobservable hypothetical entities and propose deterministic explanations of quantum mechanical phenomena. Bohmian mechanics is one of the most widely accepted hidden-variable theories. In Bohmian mechanics, the particle is guided by a wave with its initial position as the hidden variable [1]. However, non-locality was not initially included in this theory. Bohm and Vigier later modified the theory by imposing a stochastic process on the particle [2]. Nelson proposed a similar stochastic formulation of the quantum theory, in which the phase space representation of stochastic processes was used [3].

Ensemble interpretation, also called statistical interpretation, was developed based on the work of Einstein [4]. It states that the quantum state vector cannot completely describe an individual system, but only an ensemble of similarly prepared systems. The double-slits interference is one of the typical quantum phenomena which demonstrates the wave–particle duality and is reproduced by an ensemble of Bohmian trajectories [5,6,7]. However, the lack of experimental observations of quantum trajectories makes the statistical interpretation of the pilot wave remain a conceptual description. 

In recent years, weak measurement has provided a method to determine a set of average trajectories for an ensemble of particles under the minimum disturbance measure process [8,9]. The weak values obtained by weak measurement are beyond the real eigenvalues and have complex values with their imaginary parts relating to the rate of variation in the interference observation [10]. Solid evidence of the quantum trajectory was provided by the observation of average trajectories of individual photons in a double-slits interferometer that were observed through weak measurement [11]. Mahler et al. observed, in their experiment, that a particle guided by the pilot wave is non-local, even its initial position is a locally defined hidden variable in Bohmian mechanics [12]. The quantum trajectories observed through the weak measurements indicate that the quantum world is not purely probabilistic but is deterministic to a certain extent [13,14]. These experimental observations motivated us to study how to connect the deterministic ensemble to the probability distribution on the basis of the statistical language.

In the same period, complex Bohmian mechanics, quantum Hamilton mechanics, hyper-complex quantum mechanics, stochastic quantum mechanics, probability representation of quantum states, and so on, have been proposed to discuss particle dynamical behaviors in complex spacetime [15,16,17,18,19,20,21,22,23,24,25,26,27,28,29,30]. In recent years, some anomalous trajectories and complex probability expressions have been observed in some optical experiments. It is shown how an optical weak measurement of diagonal polarization can be realized by path interference between the horizontal and vertical polarization components of the input beam [31]. Zhou et al. found out that the operational trajectories of a photon in some scenarios are not continuous [32]. The interference of two 16-dimensional quantum states was observed in an experimental quantum-enhanced stochastic simulation [33]. The PT-symmetric quantum walk was experimentally realized on directed graphs with genuine photonic Fock states [34]. These experiments might provide a new classical insight to quantum mechanics. 

On the other hand, the analysis of complex random motions with trajectories described by complex probability has become a noticeable question in recent years. The local limiting theorem for the probability distribution over the random trajectories obtained from the complex-valued Ornstein–Uhlenbeck process was proven by Virchenko [35]. Methods to define the probability density in the complex coordinate and obtain the complex probability density function from the complex-valued wave function were discussed in the literature [36,37,38,39]. The quantum probability synthesized by a single chaotic complex-valued trajectory was proposed in [40]. In Jaoude’s study [41], he pointed out that any experiment can be executed on a complex probability set, which is the sum of a real set with its corresponding real probability and an imaginary set with its corresponding imaginary probability. 

In this article, we propose a new trajectory interpretation of the quantum probability in the complex plane. A quantum particle with random motion in complex space is considered in this new trajectory interpretation. The particle’s dynamic behavior is determined by the complex stochastic differential (SD) equation, in which the mean velocity is the optimal control law solved from the Hamilton–Jacobi–Bellman (HJB) equation. All physical quantities of the particle now are defined in the complex space, so we name this trajectory interpretation of quantum mechanics as complex mechanics. The Schrödinger equation and the quantum motions guided by the complex-valued wave function have been derived and described in the framework of complex mechanics in our previous study [26]. 

This paper is organized as follows. We first compare the SD equation in complex mechanics with the SD equations in Bohmian mechanics and stochastic mechanics in Section 2, where the similarities and differences of three SD equations are discussed. It is found out that the drift velocities in Bohmian mechanics and stochastic mechanics are highly related to the complex velocity in complex mechanics, as shown in Figure 1. In addition, quantum potential, which has been regarded as the main source of quantum phenomena, now plays a key role in the random motion.

An ensemble of complex quantum random trajectories (CQRTs) is used to present one possible interpretation of quantum probability. In Section 3, we apply a harmonic oscillator moving in the complex z-plane (or the x−y plane for z=x+iy) to demonstrate how to obtain the complex probability $\rho_{c}\left(t,x,y\right)$ by collecting the spatial points of the CQRTs. We further analyze the statistical spatial distribution attributed to the ensemble with different methods of collecting spatial points. We find out that the quantum probability ${\left|\Psi \left(t,x\right)\right|}^{2}$ can be reproduced from $\rho_{c}\left(t,x,y\right)$ with y=0 by collecting all the intersections of the ensemble of the CQRTs and the x-axis. On the other hand, the classical probability $\rho_{c}\left(t,x\right)$ can be reproduced from $\int \rho_{c}\left(t,x,y\right)dy$ by collecting all of the points of the CQRTs with the same x coordinate. The three statistical ways to yield the three distributions $\rho_{c}\left(t,x,0\right)$, $\rho_{c}\left(t,x\right)$, and $\rho_{c}\left(t,x,y\right)$ are shown schematically in Figure 1.

In Section 4, we acquire the joint probability $\rho_{c}\left(t,x,y\right)$ associated with the complex SD equation by solving the complex Fokker–Planck (FP) equation. It appears that the probability $\rho_{c}\left(t,x,y\right)$ obtained by the two routes in Figure 1 is identical. Section 5 presents the conclusions and discussions.

## 2. Real and Complex Random Motions

### 2.1. Real Random Motion in Bohmian Mechanics

In quantum mechanics, the time evolution of a one-dimensional quantum system is described by the Schrödinger equation: (1)iℏ∂Ψ(t,x)∂t=−ℏ22m∂2Ψ(t,x)∂x+UΨ(t,x). By expressing the wave function in the form of
(2)Ψ(t,x)=RB(t,x)eiSB(t,x)/ℏ,    RB, SB∈ℝ,Equation (1) can be separated into real and imaginary parts,
(3)∂SB∂t+12m(∂SB∂x)2+U−ℏ22m∂2RB∂x2=0,
(4)∂RB∂t=−1m∂SB∂x∂RB∂x−RB2m∂2SB∂x2,
which are known as the quantum Hamilton–Jacobi (QHJ) equation and the continuity equation, respectively. Bohm made an assumption that the particle’s motion is guided by the following law [1]:(5)pBM=mvBM=∂SB∂x.

This guidance law yields an unexpected motionless situation in eigenstates in some quantum systems. This motionless issue was resolved later by considering a random collision process [2]. The random motion of a particle in such a process can be addressed as:(6)$$dx=v_{B}dt+Ddw,$$
where $D=\sqrt{\hbar /2m}$ is the diffusion coefficient, dw is the standard Wiener process, and $v_{B}$ is the drift velocity: (7)$$v_{B}=\frac{1}{m}\frac{\partial S_{B}}{\partial x}+\frac{\hbar }{2m}\frac{\partial \left(\ln R_{B}^{2}\right)}{\partial x}.$$It can be shown that the probability density of the random displacement x solved from Equation (6) obeys the Born rule:(8)$$\rho_{B}\left(t,x\right)={\left|\Psi \left(t,x\right)\right|}^{2}={\left|R_{B}\left(t,x\right)\right|}^{2},$$
and satisfies the FP equation,
(9)$$\frac{\partial \rho_{B}\left(t,x\right)}{\partial t}=-\frac{\partial }{\partial x}\left(v_{B}\left(t,x\right)\rho_{B}\left(t,x\right)\right)+\frac{\hbar }{2m}\frac{\partial^{2}\rho_{B}\left(t,x\right)}{\partial x^{2}}.$$

The continuity Equation (4) is equivalent to the FP Equation (9) if Equations (7) and (8) are applied. The continuity equation, which presents the conservation of the probability, is the imaginary part of the QHJ equation. Here comes an interesting question: Why is the probability conservation related to the imaginary part of the QHJ equation (or the imaginary part of the Schrödinger equation)? Is it just a coincidence, or is it an ingenious arrangement made by nature? Perhaps there is some connection between the probability ${\left|\Psi \left(t,x\right)\right|}^{2}$ and the imaginary part of the Schrödinger equation, but it has not yet been discovered. We would like to put this question into discussion deeply in the framework of complex mechanics to see if we can obtain new findings about this connection. However, before doing that, we would like to introduce the other classical approach to quantum mechanics, stochastic mechanics. Then, we will compare Bohmian mechanics and stochastic mechanics to complex mechanics.

### 2.2. Real Random Motion in Stochastic Mechanics

In Nelson’s stochastic mechanics approach to quantum mechanics, he showed that the Schrödinger equation can be derived from a stochastic point of view as long as a diffusion process is imposed on the considered quantum particle [3]. The SD equation in his formalism is expressed in the following form:(10)dx=b+dt+dw(t),  x∈ℝ,
where b+(x(t),t) is the mean forward velocity, and w(t) is a Wiener process. The Wiener process dw(t) is Gaussian with zero mean, independent of the dx(s) for s≤t, and
(11)Et[dwi(t)dwj(t)]=2νδijdt,
where ν=ℏ/2m is the diffusion coefficient, and Et is the expectation value at time t. In order to derive the Schrödinger equation (1), Nelson assigned the wave function in the following form:(12)Ψ(t,x)=eRN+iSN,   RN, SN∈ℝ,
where the subscript *N* denotes Nelson’s stochastic approach to quantum mechanics (we call it stochastic mechanics for short). The mean forward velocity b+ is the sum of the current velocity vρ and the osmotic velocity uρ,
(13)b+=vρ+uρ=ℏm∂SN∂x+ℏm∂RN∂x.  Equation (10), then, can be rewritten in the form of:(14)dx=ℏm(∂SN∂x+∂RN∂x)dt+dw(t).The FP equation associated with the above SD equation is:(15)∂ρN(t,x)∂t=−∂∂x(b+ρN(t,x))+ℏ2m∂2ρN(t,x)∂x2.It can be shown that the solution of Equation (15) is Born’s probability density:(16)ρN(t,x)=|Ψ(t,x)|2=e2RN.With the help of Equations (13) and (16), the FP Equation (15) can be expressed as
(17)∂RN∂t=−ℏm∂SN∂x∂RN∂x−ℏ2m∂2SN∂x2
and can be recognized as the continuity Equation (4).

We note that the osmotic velocity $u_{\rho }$ is in connection with the probability density $\rho_{N}$ if we substitute $\rho_{N}$ for $R_{N}$, i.e., $u_{\rho }=\left(\hbar /2m\right)\left(\nabla \ln\rho_{N}\right)$. It implies that the osmotic velocity may play an important role in the trajectory interpretation to the quantum probability. The connection between the FP Equation (17) and the continuity Equation (4), i.e., the imaginary part of the Schrödinger equation, is spotted again. The question is why, in Bohmian mechanics and stochastic mechanics, the FP equation is related to the imaginary part of the Schrödinger equation, while random motion is defined on the real axis. Or, in general, why is the Schrodinger equation, which describes real motion, defined in the complex plane? 

It seems that the two classical approaches to quantum mechanics, Bohmian mechanics and stochastic mechanics, are so similar to each other; however, they are essentially different. As pointed out earlier, Bohm mechanics was built on the basis of the pilot-wave concept with the postulated guidance law, $p_{BM}=\nabla S_{B}$, and then, a modified version arose later in order to solve the motionless condition that happened in the eigenstates. This modified version indicates that particles are not only guided by the pilot wave (the wave function) but also experience a diffusion process. On the contrary, Nelson assumed that particles obey a diffusion process first and then assigned a proper wave function to particles’ mean forward velocity. The Schrödinger equation then emerges naturally from the SD equation describing the diffusion process. These two similar classical approaches certainly have something in common. For example, FP Equations (9) and (15) in Bohmian mechanics and stochastic mechanics are identical in virtue of the relationship between two wavefunction expressions,
(18)$$S_{B}\left(t,x\right)=\;\hbar S_{N}\left(t,x\right),\;\;\;\;\ln R_{B}\left(t,x\right)=R_{N}\left(t,x\right).$$The same solutions will be found by solving the two FP equations; moreover, they have the same probability density satisfying the Born rule: (19)$$\rho_{B}\left(t,x\right)=\rho_{N}\left(t,x\right)={\left|\Psi \left(t,x\right)\right|}^{2}.$$The SD equations proposed by Bohm and Nelson reconstruct Born’s probability density through random motions on the real axis. In the next subsection, we will see that a particle’s random motion in the complex plane can reflect the complex nature of the Schrödinger equation more properly and can explain the origin of Born’s probability density from complex probability.

### 2.3. Complex Random Motion in Complex Mechanics

Let us consider a random motion taken place in the complex plane z=x+iy,
(20)$$dz=u\left(t,z\right)dt+\sqrt{\nu }dw,$$
where ν represents the diffusion coefficient, u(t,z) is the drift velocity to be determined, and w is the normalized Wiener process satisfying E(w)=0 and $E\left(dw^{2}\right)=dt$. There are two displacements in Equation (20): u(t,z)dt is the drift displacement, and $\sqrt{\nu }dw$ represents the random diffusion displacement. To find the optimal drift velocity u(t,z), we minimize the cost-to-go function
(21)$$V\left(t,z\right)=\underset{u\left[t,t_{f}\right]}{\min}J\left(t,z,u\right)=E_{t,z}\left\{\int_{t}^{t_{f}}L\left(\tau .z\left(\tau \right),u\left(\tau \right)\right)d\tau \right\},$$
where $E_{t,z}\left\{\cdot \right\}$ denotes the expectation over all random trajectories starting from z(t)=z. The expectation is needed for dealing with the randomness of the cost-to-go function. It can be shown that the optimal cost-to-go function V(t,z) satisfies the stochastic HJB equation [26]:(22)$$-\frac{\partial V\left(t,z\right)}{\partial t}=\underset{u}{\min}\left\{L\left(t,z,u\right)+\frac{\partial V\left(t,z\right)}{\partial z}u+\frac{\nu }{2}\frac{\partial^{2}V\left(t,z\right)}{\partial z^{2}}\right\}.$$Under the demand of minimizing the terms inside the brace at the fixed time t and the fixed position z, the optimal command $u^{*}\left(t,z\right)$ can be determined from the condition
(23)$$\frac{\partial L\left(t,z,u\right)}{\partial u}=-\frac{\partial V\left(t,z\right)}{\partial z}.$$Therefore, the stochastic HJB equation is reduced to
(24)$$-\frac{\partial V\left(t,z\right)}{\partial t}=L\left(t,z,u^{*}\right)+\frac{\partial V\left(t,z\right)}{\partial z}u^{*}+\frac{\nu }{2}\frac{\partial^{2}V\left(t,z\right)}{\partial z^{2}}.$$One can derive the Schrödinger equation from the above stochastic HJB equation by choosing $L\left(t,z,u\right)=mu^{2}/2-U\left(z\right)$ as the Lagrangian of a particle with mass m moving in the potential U(z), and ν=−iℏ/m as the diffusion coefficient. For the given Lagrangian L(t,z,u), the optimal drift velocity $u^{*}$ can be determined from Equation (23),
(25)$$u^{*}=-\frac{1}{m}\frac{\partial V\left(t,z\right)}{\partial z}.$$The optimal cost-to-go function V(t,z) is then determined from Equation (24) with the above $u^{*}$:(26)$$\frac{\partial V}{\partial t}-\frac{1}{2m}{\left(\frac{\partial V\left(t,z\right)}{\partial z}\right)}^{2}-U-\frac{i\hbar }{2m}\frac{\partial^{2}V\left(t,z\right)}{\partial z^{2}}=0.$$In terms of the following transformations,
(27)V(t,z)=−S(t,z)=iℏlnΨ(t,z).We thus obtain two alternative forms of the HJB Equation (26)
(28)$$\frac{\partial S\left(t,z\right)}{\partial t}+\frac{1}{2m}{\left(\frac{\partial S\left(t,z\right)}{\partial z}\right)}^{2}+U-\frac{i\hbar }{2m}\frac{\partial^{2}S\left(t,z\right)}{\partial z^{2}}=0.$$
(29)$$i\hbar \frac{\partial \Psi \left(t,z\right)}{\partial t}=-\frac{\hbar^{2}}{2m}\frac{\partial^{2}\Psi \left(t,z\right)}{\partial z^{2}}+U\Psi \left(t,z\right),\;\;$$
where Equation (28) is the QHJ equation defined in the complex domain and Equation (29) is the Schrödinger equation with complex coordinate z.

It is worthy to notice that the optimal command $u^{*}$ represents the mean velocity of the random motion described by Equation (20) and is related to the wave function as
(30)$$u^{*}\left(t,z\right)=-\frac{1}{m}\frac{\partial V\left(t,z\right)}{\partial z}=\frac{1}{m}\frac{\partial S\left(t,z\right)}{\partial z}=-\frac{i\hbar }{m}\frac{\partial \left(\ln\Psi \left(t,z\right)\right)}{\partial z}.$$

We have derived the Schrödinger equation from the HJB equation in the framework of complex mechanics. The relation between the optimal cost-to-go function V(t,z) and the wave function Ψ(t,z) in Equation (27) shows that the solution of the Schrödinger equation is associated to the solution of the HJB equation in the complex z-plane. Accordingly, the Schrödinger equation and the wave function are both defined in the complex domain owing to the complex random motion described by Equations (20) and (30). The last term in Equation (28) is the complex quantum potential Q, which drives the diffusion process and can be regarded as the cause of the random motion.

By applying the optimal guidance law (30) to the SD Equation (20), we obtain the random motions in the complex z-plane
(31)$$\begin{array}{cc} dz=u^{*}dt+\sqrt{\nu }dw & =\frac{1}{m}\frac{\partial S\left(t,z\right)}{\partial z}+\sqrt{-\frac{i\hbar }{m}}dw\\ & =-\frac{i\hbar }{m}\frac{\partial \left(\ln\Psi \left(t,z\right)\right)}{\partial z}dt+\sqrt{-\frac{i\hbar }{m}}dw. \end{array}$$From Equation (31), we can see that the optimal guidance law $u^{*}$ is the drift velocity, which determines the particle’s mean motion in the complex plane. In order to compare complex mechanics to Bohmian mechanics and stochastic mechanics, we map all physical quantities from the one-dimensional complex variable z=x+iy to the two-dimensional real x−y plane. Under this mapping, the complex action function S(t,z) can be symbolically separated as
(32)$$S\left(t,z\right)=S\left(t,x+iy\right)=S_{R}\left(t,x,y\right)+iS_{I}\left(t,x,y\right).$$With the above separation and z=x+iy, we can rewrite the complex SD Equation (31) in terms of two coupled real SD equations:(33)$$dx=\frac{1}{m}\frac{\partial S_{R}\left(t,x,y\right)}{\partial x}dt-\sqrt{\frac{\hbar }{2m}}dw=v_{R}\left(t,x,y\right)dt-Ddw,\;$$
(34)$$dy=\frac{1}{m}\frac{\partial S_{I}\left(t,x,y\right)}{\partial x}dt+\sqrt{\frac{\hbar }{2m}}dw=v_{I}\left(t,x,y\right)dt+Ddw.$$The above equations can be expressed in the matrix form:(35)$$\left[\begin{array}{c} dx\\ dy \end{array}\right]=\left[\begin{array}{c} v_{R}\left(t,x,y\right)\\ v_{I}\left(t,x,y\right) \end{array}\right]dt+\left[\begin{array}{c} -D\\ D \end{array}\right]dw.$$However, in practical computation, we cannot analytically separate the complex-valued wave function into two real wave functions as described by Equation (32), since they are coupled by the complex Schrödinger equation. 

There are some similarities between the three SD equations in Bohmian mechanics, stochastic mechanics, and complex mechanics. We list some comparisons in Table 1. 

We can find relationships between the three different expressions of the wave function from Table 1: (36)$$S_{R}\left(t,x,0\right)=S_{B}\left(t,x\right)=\hbar S_{N}\left(t,x\right),$$
(37)$$S_{I}\left(t,x,0\right)=-\hbar \ln R_{B}\left(t,x\right)=-\hbar R_{N}\left(t,x\right).$$The setting of y=0 means that the domain of the complex wave function is projected from the two-dimensional z=x+iy plane onto the one-dimensional x-axis. It makes sense here since quantum mechanics, Bohmian mechanics, and stochastic mechanics all consider a physical scenario as having occurred on the real x-axis for one-dimensional quantum systems. From Equations (36) and (37) and the drift velocities in Table 1, we can find the following relationships:(38)$$v_{B}=\frac{1}{m}\frac{\partial S_{B}\left(t,x\right)}{\partial x}+\frac{\hbar }{2m}\frac{\partial \left(\ln R_{B}^{2}\left(t,x\right)\right)}{\partial x}=\frac{1}{m}\frac{\partial S_{R}\left(t,x,0\right)}{\partial x}-\frac{1}{m}\frac{\partial S_{I}\left(t,x,0\right)}{\partial x},$$
(39)$$b_{+}=v_{\rho }+u_{\rho }=\frac{\hbar }{m}\frac{\partial S_{N}\left(t,x\right)}{\partial x}+\frac{\hbar }{m}\frac{\partial R_{N}\left(t,x\right)}{\partial x}=\frac{1}{m}\frac{\partial S_{R}\left(t,x,0\right)}{\partial x}-\frac{1}{m}\frac{\partial S_{I}\left(t,x,0\right)}{\partial x}.$$The current velocity $v_{\rho }$ and the osmotic velocity $u_{\rho }$ defined in stochastic mechanics are equivalent to the real part and negative imaginary part of the complex velocity $u^{*}$ evaluated at the x-axis. What attracts our attention is that the osmotic velocities defined in Bohmian mechanics and stochastic mechanics are related to the imaginary part of the complex velocity in complex mechanics. This means that Bohmian mechanics and stochastic mechanics cannot describe quantum motions completely unless the complex domain is considered. In addition, the imaginary part of the complex velocity naturally arises from the optimization process (21), but the osmotic velocities in Bohmian mechanics and stochastic mechanics are deliberately assigned. In the following section, we will reveal how probability relates to the imaginary part of random motion. 

## 3. Extending Probability to the Complex Plane

An ensemble of CQRTs solved from the SD Equation (31) will be used in this section to obtain the probability distribution of a particle’s position in the complex plane. For a quantum harmonic oscillator with random motions in the complex plane, its dynamic behavior according to Equation (31) can be expressed as (in dimensionless form):(40)$$dz=-i\frac{\partial \left(\ln\Psi_{n}\left(t,z\right)\right)}{\partial z}dt+\sqrt{1/2}\left(-1+i\right)\xi \sqrt{dt}.$$$\Psi_{n}\left(t,z\right)$ is the complex-valued wave function of the $n^{\mathrm{th}}$ state of the harmonic oscillator
(41)$$\Psi_{n}\left(t,z\right)=C_{n}H_{n}\left(z\right)e^{-z^{2}/2}e^{-i\left(n+1/2\right)t},$$
where $H_{n}\left(z\right)$ is the Hermite polynomial and $C_{n}$ is a normalized constant. The squared magnitude of the wave function is the quantum probability according to Born’s rule,
(42)$$\rho_{B}\left(t,x\right)={\left|\Psi_{n}\left(t,x\right)\right|}^{2}={\left[C_{n}H_{n}\left(x\right)\right]}^{2}e^{-x^{2}}.$$To integrate Equation (40), we rewrite it in the following finite difference form:(43)$$z_{j+1}=z_{j}-i\left(\frac{\partial \ln\Psi_{n}\left(t,z\right)}{\partial z}\right)\Delta t+\frac{-1+i}{\sqrt{2}}\xi \sqrt{\Delta t},\;\;\;j=0,1,\cdots n,$$
where $\sqrt{\Delta t}$ stems from the standard deviation of the Wiener process dw, and ξ is a real-valued random variable with standard normal distribution N(0,1), i.e., E(ξ)=0 and $\sigma_{\xi }=1$. Equation (43) can be numerically separated into real and imaginary parts: (44)xj+1=xj+Im(∂lnΨn(tj,xj,yj)∂x)Δt−ξ2Δt,   j=0,1,⋯n,
(45)yj+1=yj−Re(∂lnΨn(tj,xj,yj)∂x)Δt+ξ2Δt,   j=0,1,⋯n,
where we note that the derivative of an analytical function $f\left(z\right)=f_{R}\left(x,y\right)+if_{I}\left(x,y\right)$ is given by
(46)$$\frac{\partial f\left(z\right)}{\partial z}=\frac{\partial f_{R}\left(x,y\right)}{\partial x}+i\frac{\partial f_{I}\left(x,y\right)}{\partial x}=\frac{\partial f_{R}\left(x,y\right)}{\partial x}-i\frac{\partial f_{R}\left(x,y\right)}{\partial y}\;=\frac{\partial f_{I}\left(x,y\right)}{\partial y}+i\frac{\partial f_{I}\left(x,y\right)}{\partial x},$$
according to the Cauchy–Riemann condition
(47)$$\frac{\partial f_{R}\left(x,y\right)}{\partial x}=\frac{\partial f_{I}\left(x,y\right)}{\partial y},\;\;\frac{\partial f_{R}\left(x,y\right)}{\partial y}=-\frac{\partial f_{I}\left(x,y\right)}{\partial x}.$$It is noted that we cannot set y=0 directly at the beginning of the iteration process to obtain the quantum mechanical or the statistic mechanical results because $x_{j}$ and $y_{j}$ are coupled to each other. We have to integrate Equations (44) and (45) simultaneously to acquire the complete random trajectories in the complex plane. 

To find the probability distribution of the trajectory of the harmonic oscillator based on Bohmian mechanics, we insert the wave function $\Psi_{1}\left(t,x\right)=\sqrt{1/2\sqrt{\pi }}2xe^{-x^{2}/2}e^{-it}$ with real coordinate x into the SD Equation (6) to yield the following SD equation (in dimensionless form): (48)$$dx=\frac{1-x^{2}}{x}dt+dw.$$We then obtain the Bohmian random trajectory by integrating Equation (48). Figure 2a illustrates the statistical distribution of an ensemble of Bohmian random trajectories, which matches the quantum probability ${\left|\Psi_{1}\left(t,x\right)\right|}^{2}$ very well. 

We next consider the harmonic oscillator with random motions in the complex plane. The equation of motion is obtained by inserting the wave function $\Psi_{1}\left(t,z\right)=\sqrt{1/2\sqrt{\pi }}2ze^{-z^{2}/2}e^{-it}$ with complex coordinate z into Equation (40),
(49)$$dz=i\frac{z^{2}-1}{z}dt+\sqrt{1/2}\left(-1+i\right)\xi \sqrt{dt}.$$From Equation (49), we can see that there are two equilibrium points, z=±1, which correspond to the two peaks of the quantum probability ${\left|\Psi_{1}\left(x\right)\right|}^{2}$, denoted by the solid red line in Figure 2b. The CQRT is obtained by integrating Equation (49) with respect to time. The probability distribution formed by the ensemble of the CQRTs is illustrated in Figure 2b. To compare this probability distribution to the quantum probability, we collect all intersections of the CQRTs and the x-axis, which is called point set A,
(50)$$\mathrm{point}\;\mathrm{set}\;A=\left\{x_{j}|\left(x_{j},0\right)\in \mathrm{CQRTs}\right\}.$$The correlation coefficient between the statistical distribution of point set A and the quantum distribution ${\left|\Psi_{1}\left(x\right)\right|}^{2}$ is up to 0.9950, as shown in Figure 2b. Hence, the statistical distribution of the ensemble of the CQRTs is consistent with results of Bohmian mechanics and quantum mechanics, when the intersections of the CQRTs and the x-axis are counted. 

Let us see what benefit we can have by extending the statistical range from the real axis to the complex plane. As is well-known, there are nodes with Ψ(x)=0 in the quantum harmonic oscillator. In our previous work [42], we solved this so-called nodal issue in the framework of complex mechanics. The statistical method we used was to collect all points of the CQRTs with the same real part x:(51)$$\mathrm{point}\;\mathrm{set}\;B=\left\{\left(x_{j},y_{k}\right)|\forall \left(x_{j},y_{k}\right)\in \mathrm{CQRTs},\;\mathrm{for}\;\mathrm{fixed}\;x_{j}\right\}$$

It can be seen that point set A is a subset of point set B with $y_{k}=0$. The statistical distribution of $x_{j}$ in point set B is demonstrated together with ${\left|\Psi_{1}\left(x\right)\right|}^{2}$ (i.e., point set A) in Figure 3. Apparently, the two distributions are distinct near the nodes $x_{\mathrm{node}}$. In terms of the trajectory interpretation, the nodes are formed with zero probability, indicating that point set A is empty when it is evaluated at $x_{j}=x_{\mathrm{node}}$, as shown by the red curve in Figure 3. However, point set B is not empty when evaluated at $x_{j}=x_{\mathrm{node}}$ due to the inclusion of the extra points $\left(x_{\mathrm{node}},y_{k}\right)$ with non-zero imaginary part $y_{k}$. The two different ways of collecting the data points cause the discrepancy between the distributions of the two point sets near the nodes. The significant contribution made by including extra complex points in point set B is that the statistical distribution of $x_{j}$ in point set B converges to the classical probability distribution as the quantum number n is large, as shown in Figure 4, where point set B is generated by the CQRTs solved from Equations (44) and (45) with n=60. On the contrary, if point set A is generated from the same ensemble of CQRTs, its statistical distribution, i.e., the distribution of ${\left|\Psi_{60}\left(x\right)\right|}^{2}$, shows the existence of 60 nodes located along the real axis, which is remarkably different from the classical distribution, as shown by the green curve in Figure 4. Therefore, the extension of quantum probability to the complex plane is crucial to the applicability of the correspondence principle. 

It is obvious to see that the imaginary coordinates of the points are what make the probability distributions different. This finding reflects that the imaginary part of the random motion is directly in connection with the probability distribution. Furthermore, the imaginary part of the energy conservation (the imaginary part of the QHJ equation) constrains the imaginary part of the particle’s motion. It is very difficult to find this connection in Bohmian mechanics and stochastic mechanics since only the real part of the random motion is considered. We can safely say that the whole information of the complex Schrödinger equation can only be obtained by considering the complex domain. The description of the quantum world cannot be complete unless the actual domain and observation domain are on equal footing. 

## 4. Solving Real and Complex Probability from the Fokker–Planck Equation

In this section, we will confirm the correctness of the statistical distributions of point set B by comparing it with the solution of the FP equation. We will solve the FP equations in Bohmian mechanics and complex mechanics for a harmonic oscillator in the n=1 state. The general form of the n-dimensional SD equation reads
(52)$$\left[\begin{array}{c} dx_{1}\\ dx_{2}\\ \vdots \\ dx_{n} \end{array}\right]=\left[\begin{array}{c} v_{1}\left(t,x\right)\\ v_{2}\left(t,x\right)\\ \vdots \\ v_{n}\left(t,x\right) \end{array}\right]dt+\left[\begin{array}{ccc} \sigma_{11} & \sigma_{12} & \cdots \\ \sigma_{21} & \sigma_{22} & \cdots \\ \vdots & \vdots & \cdots \\ \sigma_{n1} & \sigma_{n2} & \cdots \end{array}\begin{array}{c} \sigma_{1m}\\ \sigma_{2m}\\ \vdots \\ \sigma_{nm} \end{array}\right]\left[\begin{array}{c} dw_{1}\\ dw_{2}\\ \vdots \\ dw_{m} \end{array}\right],$$
where $x={\left[x_{1}\;x_{2}\cdots x_{n}\right]}^{T}$ denotes the random displacement in the n-dimensional space, $v_{i}$ is the diffusion velocity (i=1,2,⋯,n), and $dw_{i}$ is the Wiener process (i=1,2,⋯,m). The joint probability density $\rho \left(t,x\right)=\rho \left(t,x_{1},x_{2},\cdots ,x_{n}\right)$ describing the spatial distribution of $x_{i}$ satisfies the n-dimensional FP equation
(53)$$\frac{\partial \rho \left(t,x\right)}{\partial t}=-\overset{n}{\underset{i=1}{\sum }}\partial_{x_{i}}\left[v_{i}\left(t,x\right)\rho \left(t,x\right)\right]+\frac{1}{2}\overset{n}{\underset{i=1}{\sum }}\overset{n}{\underset{j=1}{\sum }}\partial_{x_{i}}\partial_{x_{j}}\left[D_{ij}\left(t,x\right)\rho \left(t,x\right)\right],$$
where
(54)$$D_{ij}\left(t,x\right)=\overset{m}{\underset{k=1}{\sum }}\sigma_{ik}\left(t,x\right)\sigma_{jk}\left(t,x\right).$$The two-dimensional FP equation corresponds to Equation (35) can be derived as
(55)$$\frac{\partial \rho_{c}}{\partial t}=-\frac{\partial \left(v_{R}\rho_{c}\right)}{\partial x}-\frac{\partial \left(v_{I}\rho_{c}\right)}{\partial y}+\frac{D^{2}}{2}\left(\frac{\partial^{2}\rho_{c}}{\partial x^{2}}-2\frac{\partial^{2}\rho_{c}}{\partial x\partial y}+\frac{\partial^{2}\rho_{c}}{\partial y^{2}}\right)$$
by applying n=2, m=1, $D_{11}=\sigma_{11}^{2}=D^{2}$, $D_{12}=\sigma_{11}\sigma_{21}=-D^{2}$, $D_{21}=\sigma_{21}\sigma_{11}=-D^{2}$, and $D_{22}=\sigma_{21}^{2}=D^{2}$ to Equation (53), in which $\rho_{c}\left(t,x,y\right)$ is the probability of finding the particle in the complex plane at position z=x+iy and time t.

The finite difference method is the most common method to solve the partial differential equation by discretizing the spatial and time domains. Firstly, we will verify the correctness of our finite difference algorithm by solving the FP equation for the Duffing oscillator and comparing it with the exact solution. The two-dimensional random motion (X(t),Y(t)) for the Duffing oscillator is governed by the following SD equations:(56)$$\dot{Y}+2\alpha Y+\beta X+\gamma X^{3}=\sigma W\left(t\right),\;\;Y=\dot{X},$$
where W(t) is the random Brownian increment, and α, β, γ, and σ are given constants. An exact solution of the joint probability ρ(X,Y) can be found as
(57)$$\rho \left(X,Y\right)=C\exp\left\{-\frac{2\alpha }{\sigma^{2}}\left(Y^{2}+\beta X^{2}+\frac{\gamma }{2}X^{4}\right)\right\},$$
where C is a normalized constant. The corresponding FP equation of Equation (56) is
(58)$$\frac{\partial \rho }{\partial t}=\frac{\partial \left(2\alpha Y+\beta X+\gamma X^{3}\right)\rho }{\partial Y}-Y\frac{\partial \rho }{\partial X}+\frac{\sigma^{2}}{2}\frac{\partial^{2}\rho }{\partial Y^{2}}.$$The initial distribution is chosen as the two-dimensional Gaussian distribution:(59)$$\rho \left(X^{\left(0\right)},\;Y^{\left(0\right)}\right)=\frac{1}{2\pi \theta_{1}\theta_{2}}\exp\left[-{\left(\frac{X^{\left(0\right)}-\mu_{1}}{2\theta_{1}}\right)}^{2}-{\left(\frac{Y^{\left(0\right)}-\mu_{2}}{2\theta_{2}}\right)}^{2}\right].$$

Figure 5 displays the exact solution (57) and the finite difference solution to Equation (58). The consistent result indicates that our finite difference algorithm works very well.

Next, we apply the finite difference algorithm to solve the FP equation corresponding to the Bohmian SD Equation (9) for the quantum harmonic oscillator in the n=1 state. The statistical distribution $\rho_{B}\left(t,x\right)$ of the Bohmian random trajectory satisfies the FP equation:(60)$$\frac{\partial \rho_{B}\left(t,x\right)}{\partial t}=\frac{x^{2}+1}{x^{2}}\rho_{B}\left(t,x\right)+\frac{x^{2}-1}{x^{2}}\frac{\partial \rho_{B}\left(t,x\right)}{\partial x}+\frac{1}{2}\frac{\partial^{2}\rho_{B}\left(t,x\right)}{\partial x^{2}},$$
whose finite difference model reads
(61)$$\begin{array}{l} \rho_{B,j}^{m}=\rho_{B,j}^{m-1}\\ \;\;\;\;\;+\Delta t\left(\frac{x_{j}^{2}+1}{x_{j}^{2}}\rho_{B,j}^{m-1}+\frac{x_{j}^{2}-1}{x_{j}^{2}}\frac{\rho_{B,j+1}^{m-1}-\rho_{B,j-1}^{m-1}}{2\Delta x}+\frac{1}{2}\frac{\rho_{B,j+1}^{m-1}-2\rho_{B,j}^{m-1}+\rho_{B,j-1}^{m-1}}{{\left(\Delta x\right)}^{2}}\right). \end{array}$$

Figure 6 shows that the numerical solution to Equation (61) is in good agreement with the quantum probability ${\left|\Psi_{1}\left(x\right)\right|}^{2}$. This result indicates that the quantum probability ${\left|\Psi_{1}\left(x\right)\right|}^{2}$ can be exactly synthesized by the real random motions satisfying the Bohmian SD Equation (9); or, equivalently, the quantum probability ${\left|\Psi_{1}\left(x\right)\right|}^{2}$ is the solution $\rho_{B}\left(t,x\right)$ to the FP Equation (60).

We now extend the random motion of the harmonic oscillator in the n=1 state to the complex plane z=x+iy. The related SD equations are: (62)$$dx=v_{R}dt-Ddw=\frac{-x^{2}y-y^{3}-y}{x^{2}+y^{2}}dt-\frac{1}{\sqrt{2}}dw,$$
(63)$$dy=v_{I}dt+Ddw=\frac{x^{3}+xy^{2}-x}{x^{2}+y^{2}}dt+\frac{1}{\sqrt{2}}dw.$$According to Equation (55), the FP equation for the joint probability $\rho_{c}\left(t,x,y\right)$ of the above SD equations reads
(64)$$\frac{\partial \rho_{c}\left(t,x,y\right)}{\partial t}=\frac{-4xy}{{\left(x^{2}+y^{2}\right)}^{2}}\rho_{c}+\frac{x^{2}y+y^{3}+y}{x^{2}+y^{2}}\frac{\partial \rho_{c}}{\partial x}+\frac{x-xy^{2}-x^{3}}{x^{2}+y^{2}}\frac{\partial \rho_{c}}{\partial y}\;+\frac{1}{4}\left(\frac{\partial^{2}\rho_{c}}{\partial x^{2}}-2\frac{\partial^{2}\rho_{c}}{\partial x\partial y}+\frac{\partial^{2}\rho_{c}}{\partial y^{2}}\right),$$ where $\;\rho_{c}\left(t,x,y\right)$ is the probability of finding a particle in the complex plane. The finite difference model of Equation (64) is given by
$$\rho_{j,k}^{m}=\rho_{j,k}^{m-1}+[\frac{-4x_{j}y_{k}}{{\left(x_{j}^{2}+y_{k}^{2}\right)}^{2}}\rho_{j,k}^{m-1}+\frac{x_{j}^{2}y_{k}+y_{k}^{3}+y_{k}}{x_{j}^{2}+y_{k}^{2}}\frac{\rho_{j+1,k}^{m-1}-\rho_{j-1,k}^{m-1}}{2\Delta x}+\;\frac{x_{j}-x_{j}y_{k}^{2}-x_{j}^{3}}{x_{j}^{2}+y_{k}^{2}}\frac{\rho_{j,k+1}^{m-1}-\rho_{j,k-1}^{m-1}}{2\Delta y}-\frac{\rho_{j+1,k+1}^{m-1}-\rho_{j+1,k-1}^{m-1}-\rho_{j-1,k+1}^{m-1}-\rho_{j-1,k-1}^{m-1}}{8\Delta x\Delta y}$$
(65)$$+\frac{\rho_{j+1,k}^{m-1}-2\rho_{j,k}^{m-1}+\rho_{j-1,k}^{m-1}}{4{\left(\Delta x\right)}^{2}}+\frac{\rho_{j,k+1}^{m-1}-2\rho_{j,k}^{m-1}+\rho_{j,k-1}^{m-1}}{4{\left(\Delta y\right)}^{2}}]\Delta t.$$The numerical result is shown by the blue dashed curve in Figure 7a, where we can see that the node at x=0 for ${\left|\Psi_{1}\left(x\right)\right|}^{2}$ does not appear in the solution $\rho_{c}\left(t,x,y\right)$ to Equation (64). This means that the probability of finding a particle at the node is not zero. It is because a particle moving in the complex plane can bypass the node $\left(x_{\mathrm{node}},0\right)$ through another point $\left(x_{\mathrm{node}},y_{k}\right)$ with non-zero imaginary component $y_{k}$. Figure 7b shows that the numerical solution to the complex FP Equation (64) is consistent with the statistical distribution of point set B, which is generated by the CQRTs solved from the complex SD Equations (62) and (63). Both curves in Figure 7b show a non-zero probability of finding a particle at the node x=0.

A similar trend occurs in the n=3 state, as shown in Figure 8. We can see that the statistical distribution of an ensemble of CQRTs (the black dotted line in Figure 8a) is identical to the probability density $\rho_{c}\left(t,x,y\right)$ solved from the complex FP equation (the blue dashed line in Figure 8b), and both curves deviate from the quantum probability ${\left|\Psi_{3}\left(x\right)\right|}^{2}$ (the red solid line) near the nodes of $\Psi_{3}\left(x\right)$. This result once again shows that the occurrence of nodes is purely due to the fact that the movement of particles is restricted to the real axis by the requirement of quantum mechanics.

So far, our attention to quantum probability has focused on the real axis. In complex mechanics, quantum particles move randomly in the complex plane, and the probability distribution of their locations must be expressed in the complex x−y plane instead of the real axis. Figure 9 illustrates the probability distribution $\rho_{c}\left(t,x,y\right)$ over the complex x−y plane solved from the complex FP Equation (64). The inset in the figure shows the contour plot of $\rho_{c}\left(t,x,y\right)$, from which we can see that the joint probability $\rho_{c}\left(t,x,y\right)$ reaches peaks around the points (x,y)=(1,1) and (−1,−1) and declines to the node at (x,y)=(0,0). The 3D plot of $\rho_{c}\left(t,x,y\right)$ manifests that when $\sqrt{x^{2}+y^{2}}>3$, $\rho_{c}\left(t,x,y\right)$ approaches zero, which means that a particle in the n=1 state is bound along the real axis as well as the imaginary axis and will not be too far from the origin. By contrast, quantum probability ${\left|\Psi_{1}\left(x\right)\right|}^{2}$ only concerns particles on the real axis, so it only provides the probability distribution along the x-axis.

Just as $\rho_{B}\left(t,x\right)={\left|\Psi \left(t,x\right)\right|}^{2}$ gives the probability of finding a particle at position x on the real axis, the joint probability density $\rho_{c}\left(t,x,y\right)$ gives the probability that a particle appears at the position z=x+iy in the complex plane. The $\rho_{c}\left(t,x,y\right)$ illustrated in Figure 10 shows a more complicated probability distribution over the complex plane as the particle moves in the n=3 state. If we sum the probability $\rho_{c}\left(t,x,y\right)$ for all the values of y along a vertical line $x=x_{0}$ in the complex plane (i.e., point set B), the 1D probability distribution $\rho_{c}\left(t,x_{0}\right)$ will recover the result of Figure 7. Mathematically, $\rho_{c}\left(t,x,y\right)$ and $\rho_{c}\left(t,x\right)$ have the following relation
(66)$$\rho_{c}\left(t,x\right)=\int_{-\infty }^{+\infty }\rho_{c}\left(t,x,y\right)dy.$$The evolution from Figure 8 (Figure 7) to Figure 10 (Figure 9), i.e., from $\rho_{c}\left(t,x\right)$ to $\rho_{c}\left(t,x,y\right)$, is just the process by which we extend the definition of the quantum probability from the real axis to the complex plane.

Complex probability is a puzzle in complex-extended quantum mechanics. It is so obscure and abstract and even hard to define since the probability must be a positive number. One of the most convincing solutions is to directly extend Born’s definition of probability to complex coordinates. Born’s probability density $\rho_{B}\left(t,x\right)={\left|\Psi \left(t,x\right)\right|}^{2}$ is originally defined on the real axis. After replacing the real coordinate x with the complex coordinate z=x+iy, we have a joint probability density $\rho_{B}\left(t,x,y\right)={\left|\Psi \left(t,z\right)\right|}^{2}={\left|\Psi \left(t,x+iy\right)\right|}^{2}$. Since ${\left|\Psi \left(t,x\right)\right|}^{2}dx$ correctly predicts the probability of finding a quantum particle in the interval between x and x+dx at time t, it is natural to expect that ${\left|\Psi \left(t,\;x+iy\right)\right|}^{2}dxdy$ can provide the probability of finding a quantum particle inside the infinitesimal region spanned by dx and dy in the complex plane z=x+iy. However, such an expectation ultimately falls short, because the square-integrable condition imposed on Ψ(t,x) can only guarantee that ${\left|\Psi \left(t,x\right)\right|}^{2}$ is a qualified probability density—it cannot guarantee that ${\left|\Psi \left(t,x+iy\right)\right|}^{2}$ is also qualified. To show that ${\left|\Psi \left(t,z\right)\right|}^{2}$ is not a qualified probability measure in the complex domain, the magnitude plot of ${\left|\Psi_{1}\left(t,z\right)\right|}^{2}$ over the complex plane z=x+iy is shown in Figure 11, where we can observe ${\left|\Psi_{1}\left(t,z\right)\right|}^{2}\to 0$ as |x|→∞, and ${\left|\Psi_{1}\left(t,z\right)\right|}^{2}\to \infty$ as |y|→∞. The observed features of ${\left|\Psi_{1}\left(t,z\right)\right|}^{2}$ indicate that ${\left|\Psi_{1}\left(t,x+iy\right)\right|}^{2}$ cannot be used as a probability measure along the imaginary axis y. The correct probability density $\rho_{c}\left(t,x,y\right)$ describing a particle’s motion with n=1 in the complex plane is shown in Figure 9, which is solved from the complex FP Equation (64) and is significantly different from $\rho_{B}\left(t,x,y\right)={\left|\Psi_{1}\left(t,x+iy\right)\right|}^{2}$.

In this paper, we follow de Broglie’s original intention and regard the wave function Ψ(t,z) as a guided wave that guides a particle’s motion in the complex plane rather than as a probability density ${\left|\Psi \left(t,z\right)\right|}^{2}$. When we extend the quantum probability from the real axis to the complex plane, the complex SD equation (31) plays a key role, because the CQRTs solved from it completely determine the probability distribution of the particles in the complex plane including the real axis. The complex SD equation (31) determined by the wave function Ψ(t,z) can be used to describe the random motion of particles in the complex plane. Along the random trajectory of the particles, we recorded the number of times the particles appear at different positions in the complex plane and then obtained point set A and point set B. From the distribution in point set A, we reconstructed the probability of particles appearing on the real axis and confirmed that the obtained probability is identical to the Born probability $\rho_{B}\left(t,x\right)={\left|\Psi \left(t,x\right)\right|}^{2}$. On the other hand, from the distribution in point set B, we obtained the probability that the real-part position of the particle is equal to $x_{j}$ and proved that when the quantum number increases, the probability distribution of $x_{j}$ obtained from point set B gradually approaches classical probability distribution, as shown in Figure 4.

## 5. Conclusions

The quantum world with the most mysterious phenomena is described by the weirdest theory, quantum mechanics. Probability throughout quantum mechanics is the result of empirical observations, which is so different to our familiar classical theory and is also counterintuitive. The trajectory interpretation of quantum mechanics provides a possible concrete meaning to probability. In this article, we introduced and compared three trajectory interpretations on the basis of Bohmian mechanics, stochastic mechanics, and complex mechanics. The first two mechanics consider that particles are moving randomly along the real x-axis for one-dimensional quantum systems, while complex mechanics considers random motion in the complex plane z=x+iy. We found out that the osmotic velocities defined in Bohmian mechanics and stochastic mechanics are related to the imaginary part of the complex velocity in complex mechanics. This relation reflects that the random motion along the imaginary y-axis is responsible for the osmotic motion, and only by considering a particle’s motion in the complex plane can we obtain its complete information.

Our research reveals that there is no contradiction if the quantum probability, which is originally defined on the real axis, is extended from the real axis to the complex plane. Moreover, the complex domain extension can even help us to capture the origin of the actual probability in the microscopic world. From particles’ random motion in the complex plane, we found the reason why quantum probability is defined on the real axis. It turns out that a particle’s position predicted by quantum mechanics is the intersection of the particle’s complex trajectory and the real axis. By solving the complex SD equation, we collected all the intersection points of the particle’s complex trajectory and the real axis and calculated the probability distribution of these intersection points on the real axis, and we found that the obtained probability is exactly the same as the Born quantum probability. 

On the other hand, the quantum probability established by the intersections of the complex random trajectories and the vertical line $x=x_{0}=\mathrm{constant}$ overcomes the classical contradictory condition of the node existence of the harmonic oscillator. We pointed out that classical probability $\rho_{c}\left(t,x\right)$ is actually the result of integrating complex probability $\rho_{c}\left(t,x,y\right)$ with respect to the imaginary part y of a particle’s position, as shown in Equation (66). The classical probability obtained in this way is not zero even at the node, i.e., $\rho_{c}\left(t,x_{\mathrm{node}}\right)\neq 0$. Through the random trajectory of a particle in the complex plane, we counted the probability distribution of the particle’s position in the complex plane to establish $\rho_{c}\left(t,x,y\right)$, and then obtained the classical probability $\rho_{c}\left(t,x\right)$ through the integral operation in Equation (66) (for discrete data, it is an additional operation). When the quantum number is large, we confirm that the marginal probability $\rho_{c}\left(t,x\right)$ obtained by integrating $\rho_{c}\left(t,x,y\right)$ with respect to y is the probability defined by classical mechanics.

In conclusion, we used complex random motion to integrate quantum probability, classical probability, and complex probability. It was demonstrated that the three probability measures can all be established by the distribution of a particle’s random positions in the complex plane, and the difference between them is only in the way of counting the particle’s positions. As shown in Figure 1, the quantum probability $\rho_{B}\left(t,x_{0}\right)$ counts the number of times that the complex trajectories intersect the real axis at a certain point $\left(x_{0},0\right)$; the classical probability $\rho_{c}\left(t,x_{0}\right)$ counts the number of times that the complex trajectories intersect a certain vertical line ($x_{0},y$); and the complex probability $\rho_{c}\left(t,x_{0},y_{0}\right)$ counts the number of times the complex trajectories pass a certain fixed point $\left(x_{0},y_{0}\right)$ in the complex plane. After we establish the complex probability $\rho_{c}\left(t,x,y\right)$, we can integrate $\rho_{c}\left(t,x,y\right)$ with respect to y to obtain the classical probability $\rho_{c}\left(t,x\right)$, and we can evaluate $\rho_{c}\left(t,x,y\right)$ at y=0 to obtain the quantum probability $\rho_{c}\left(t,x,0\right)={\left|\Psi \left(t,x\right)\right|}^{2}$. Only by defining probability in the complex plane can we see the difference between the quantum probability $\rho_{c}\left(t,x,0\right)$ and the classical probability $\rho_{c}\left(t,x\right)$.

There are already some experiments supporting the assumption of quantum motion in the complex plane, and we believe that there will be more evidence to disclose the complex properties of the quantum world in the near future.

## Figures and Tables

**Figure 1 entropy-23-00210-f001:**
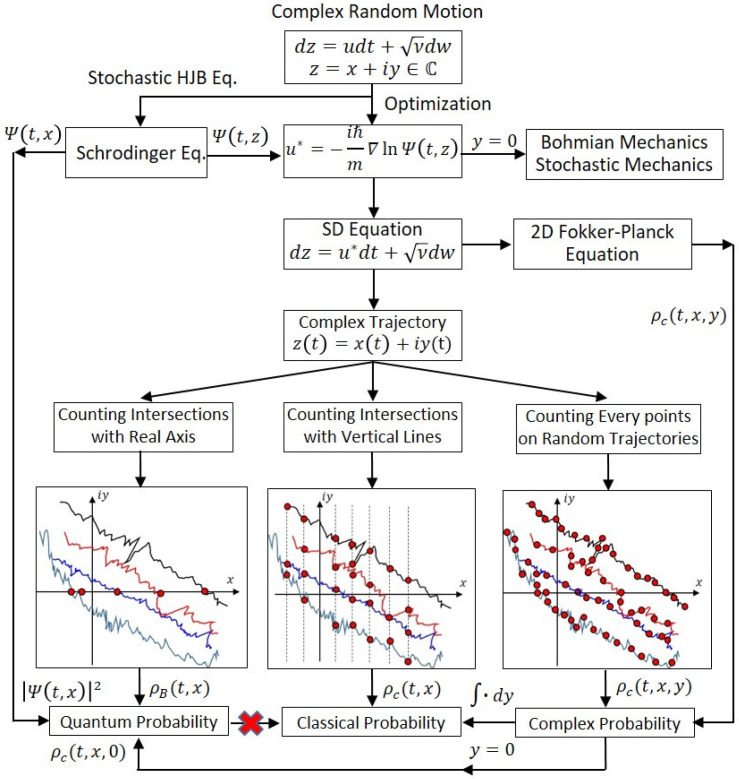
A chart summarizing the main findings of this article by revealing the relationships between quantum probability, classical probability, and complex probability, based on complex random motion. The special probability distribution $\rho_{c}\left(t,x,0\right)$, which represents the statistical distribution of the intersections of the ensemble of the complex quantum random trajectories (CQRTs) and the real axis, is shown to reproduce the quantum probability ${\left|\Psi \left(t,x\right)\right|}^{2}$. On the other hand, the marginal distribution $\rho_{c}\left(t,x\right)$ obtained by integrating $\rho_{c}\left(t,x,y\right)$ with respect to the imaginary part y can reproduce the classical probability.

**Figure 2 entropy-23-00210-f002:**
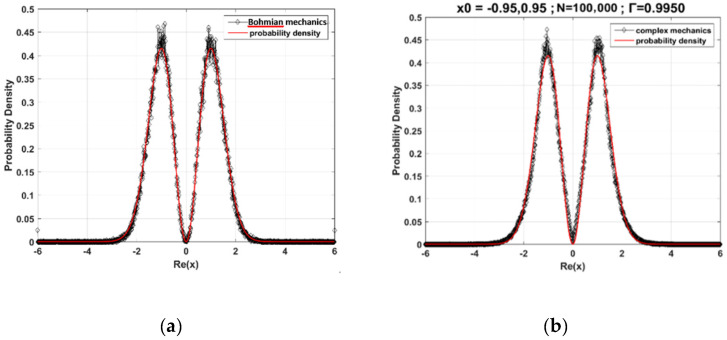
(**a**) The statistical distribution of an ensemble of Bohmian random trajectories for n=1 state of the harmonic oscillator. The solid red line represents the quantum probability distribution and the black dotted line denotes the statistical distribution of the ensemble given by Bohmian mechanics. (**b**) The statistical distribution of an ensemble of complex quantum random trajectories (CQRTs) for n=1 state is given by collecting the intersections of CQRTs and the real axis. The initial positions are $x_{0}=\pm 0.95$ and there are 100,000 trajectories. The solid red line is the quantum probability distribution and the black dotted line is the statistical distribution of the ensemble given by complex mechanics. The correlation coefficient related to the quantum mechanical probability distribution is Γ=0.9950.

**Figure 3 entropy-23-00210-f003:**
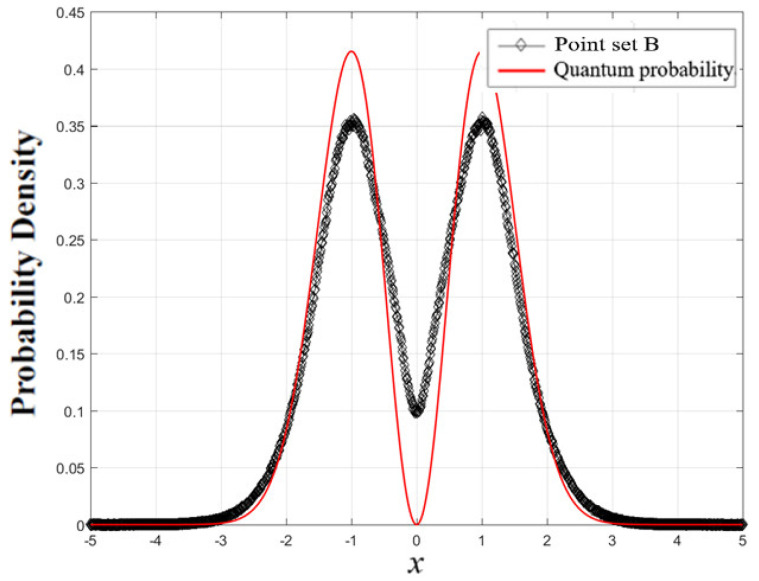
The statistical distribution of an ensemble of intersection points and projection points on the real axis of the CQRTs for n=1 state. The red solid line is the quantum probability distribution and the black dotted line is the statistical distribution of point set B [42].

**Figure 4 entropy-23-00210-f004:**
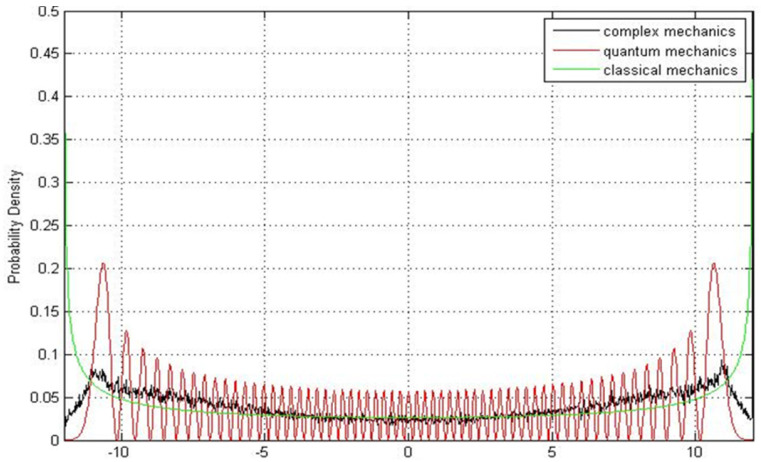
The probability distribution (black curve) constructed from an ensemble of CQRTs for a harmonic oscillator with n=60 approaches classical distribution (green curve). The quantum probability ${\left|\Psi_{60}\left(t,x\right)\right|}^{2}$ denoted by the red curve has 60 nodes located along the x-axis, which is remarkably different from the classical distribution.

**Figure 5 entropy-23-00210-f005:**
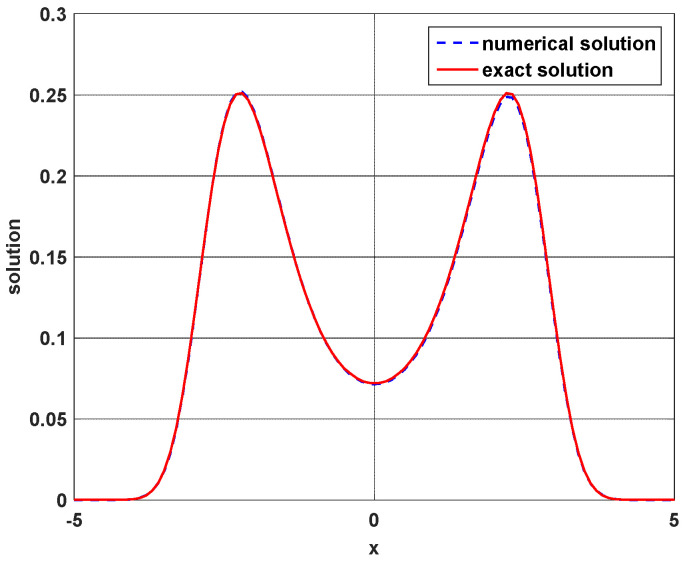
Compatible results are given by the exact solution (red solid line) and the numerical solution (blue dashed line) of the Fokker–Planck (FP) equation for the one-dimensional Duffing oscillator. The parameter settings in the simulation are: α=0.25, β=−1, γ=0.2, σ=1, $\theta_{1}=\theta_{2}=0.1$, $\mu_{1}=-2$, $\mu_{2}=-1.8$, dX=dY=0.05, and dt=0.0001.

**Figure 6 entropy-23-00210-f006:**
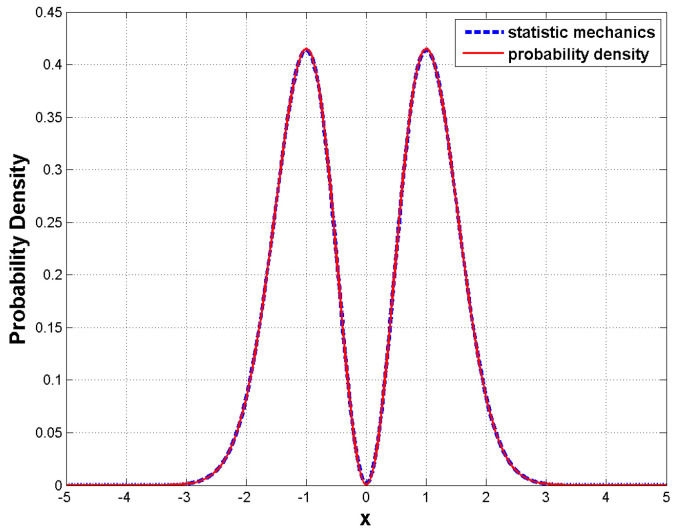
The numerical solution of the FP equation (blue dashed line) of the harmonic oscillator in the n=1 state in Bohmian mechanics is consistent with the quantum probability distribution (red solid line). The initial probability is assigned as $\rho_{B}\left(0,x\right)=2x^{2}e^{-x^{2}}/\sqrt{\pi }$ and the boundary conditions are $\rho_{B}\left(t,-5\right)=\rho_{B}\left(t,5\right)=0.$

**Figure 7 entropy-23-00210-f007:**
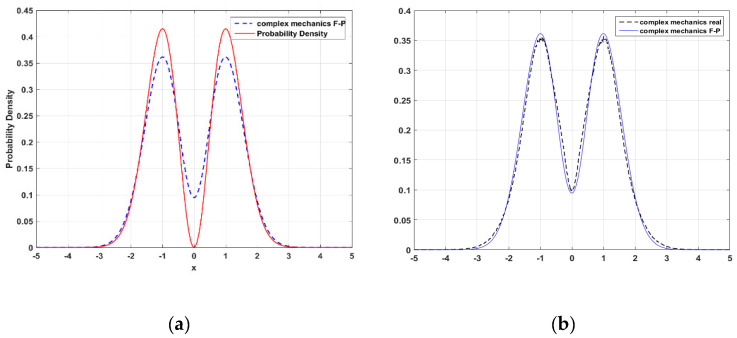
(**a**) Comparison of the numerical solution of the complex FP equation (blue dashed line) to the quantum probability ${\left|\Psi_{1}\left(t,x\right)\right|}^{2}$ (red solid line). (**b**) Comparison of the numerical solution of the complex FP equation (blue solid line) with the statistical distribution of point set B generated by the CQRTs for the n=1 state (black dashed line). The initial probability is assigned as $\rho_{c}\left(0,x,y\right)=2\left(x^{2}+y^{2}\right)e^{-\left(x^{2}+y^{2}\right)}/\sqrt{\pi }$. The boundary conditions are $\rho_{c}\left(t,-5,y\right)=\rho_{c}\left(t,5,y\right)=0,$ and $\rho_{c}\left(t,x,-5\right)=\rho_{c}\left(t,x,5\right)=0.$

**Figure 8 entropy-23-00210-f008:**
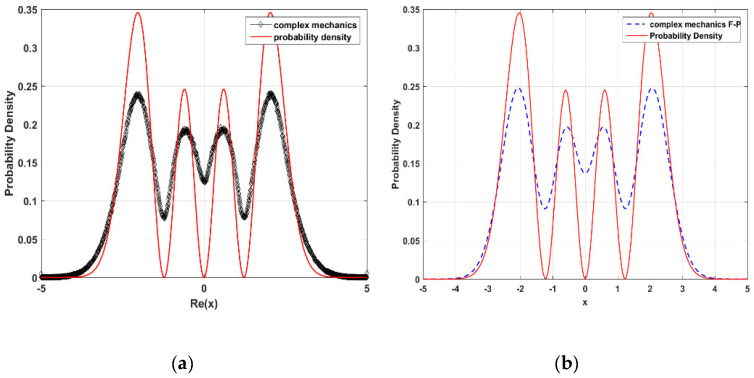
(**a**) The red solid line is the quantum probability ${\left|\Psi_{3}\left(t,x\right)\right|}^{2}$ and the black dotted line is the statistical distribution of point set B generated by CQRTs for n=3 state. (**b**) Comparison of the solution of the complex FP equation (blue dashed line) to the quantum probability ${\left|\Psi_{3}\left(t,x\right)\right|}^{2}$ (red solid line). The initial probability is assigned as $\rho_{c}\left(0,x,y\right)=\left(e^{-\left(x^{2}+y^{2}\right)}/3\sqrt{\pi }\right)\left[4\left(x^{6}+y^{6}\right)-12\left(x^{4}+y^{4}\right)+9\left(x^{2}+y^{2}\right)\right]$. The boundary conditions are $\rho_{c}\left(t,-5,y\right)=\rho \left(t,5,y\right)=0$ and $\rho_{c}\left(t,x,-5\right)$
=ρ(t,x,5)=0.

**Figure 9 entropy-23-00210-f009:**
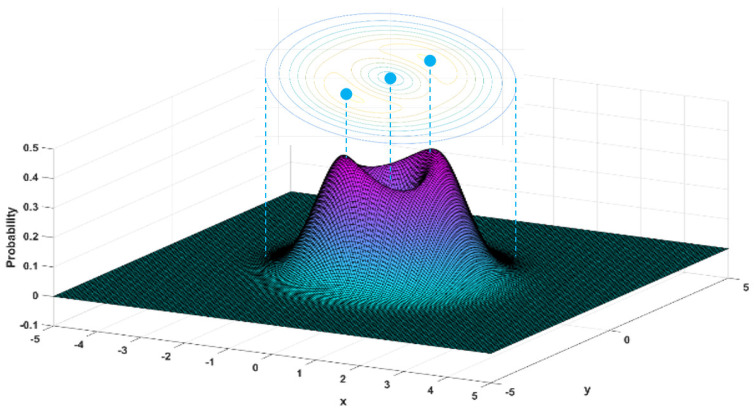
The 3D distribution and contour of complex probability $\rho_{c}\left(t,x,y\right)$ for n=1 state are obtained by solving the complex FP equation in the x−y plane.

**Figure 10 entropy-23-00210-f010:**
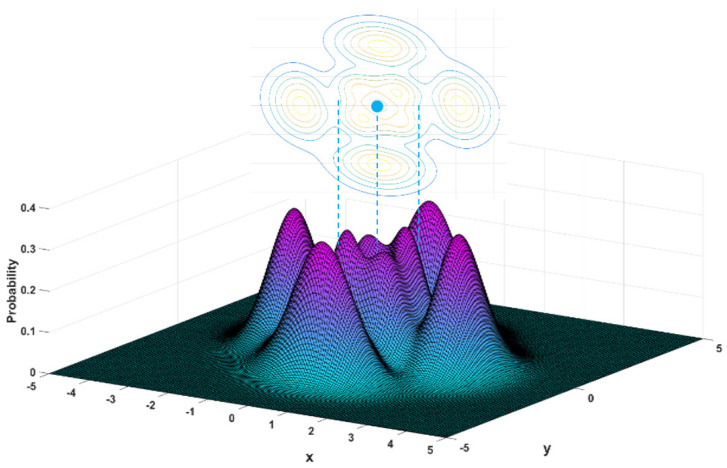
The 3D distribution and contours of the complex probability $\rho_{c}\left(t,x,y\right)$ for n=3 state are obtained by solving the complex FP equation in the x−y plane.

**Figure 11 entropy-23-00210-f011:**
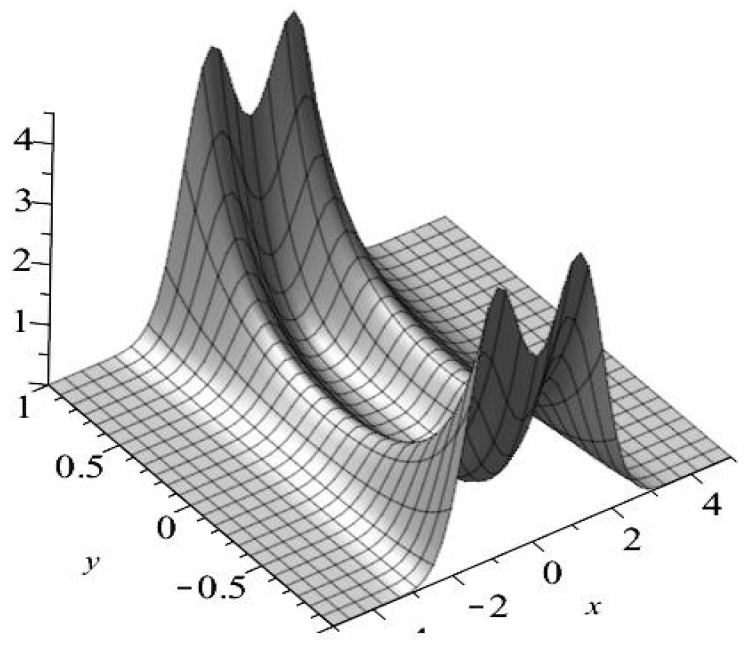
Magnitude plot of ${\left|\Psi_{1}\left(z\right)\right|}^{2}$ over the complex plane z=x+iy shows ${\left|\Psi_{1}\left(z\right)\right|}^{2}\to 0$ as |x|→∞, and ${\left|\Psi_{1}\left(z\right)\right|}^{2}\to \infty$ as |y|→∞, which means that ${\left|\Psi_{1}\left(z\right)\right|}^{2}$ cannot be used as a probability measure along the imaginary axis.

**Table 1 entropy-23-00210-t001:** Comparisons of the stochastic differential (SD) equations and some related terms in three mechanics.

	Bohmian Mechanics	Stochastic Mechanics	Complex Mechanics
Domain	real x-axis	real x-axis	complex z=x+iy plane
Wave function	Ψ(t,x)=RBeiSB/ℏ	Ψ(t,x)=eRN+iSN	Ψ(t,z)=eiS(t,z)/ℏ
Drift velocity	vB=1m∂SB∂x+ℏ2m∂(lnRB2)∂x	b+=ℏm∂SN∂x+ℏm∂RN∂x	u*=1m∂S(t,z)∂z=−iℏm∂(lnΨ(t,z))∂z
SD equation	dx=vBdt+Ddw, x∈ℝ	dx=b+dt+dw, x∈ℝ	dz=u*dt+νdw, z∈ℂ
PDF	ρB(t,x)=|Ψ(t,x)|2 Satisfying Schrödinger Equation	ρN(t,x)=|Ψ(t,x)|2 Satisfying Schrödinger Equation	ρc(t,x,y)≠|Ψ(t,x+iy)|2 Satisfying 2D Fokker–Planck Equation

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
