# Peer review of "Extending Quantum Probability from Real Axis to Complex Plane"

_entropy, 2021, doi:10.3390/e23020210_

Round 1
Reviewer 1 Report
Please see attached file.

Author Response
Reply to reviewer 1 is in the upload file.

Reviewer 2 Report
Reviewer report:
This paper puts forward a stochastic model of a particle moving in a one dimensional complex space from which both the quantum and the classical probabilities are derived after imposing proper conditions. The idea is pretty interesting and the arguments are basically well presented along with detailed numerical results in the example of the harmonic oscillator. Thus I believe that this paper can be published provided that the authors implement revisions in the following points.
1) Despite its crucial importance and extensive use throughout the paper, the probability distribution "rho_c(t,x,y)" does not seem to be defined explicitly anywhere in the text. The authors must provide the definition by providing an equation to clarify what it is.
2) Related to the above, the authors should give the reason why the quantity "rho_c(t,x,y)" qualifies as probability distribution. Is it simply because the marginal distribution defined in (59) becomes a probability distribution over the space x?
3) There are portions which look somewhat sloppy in the text. For instance, the same notation for the velocity v_B is used both in (5) and (7) but obviously they are contradictory to each other. Also, in (41) a derivative (in the difference form) appears to be missing in the second term of the r.h.s. containing Psi_n(t,z). These should be corrected appropriately.
4) How can this prposed stochastic model defined by making the one dimensional space complex be generalized to the three spatial dimensions where we live? The authors are advised to comment on this in the last concluding section.
Author Response
Reply to reviewer 2 is in the upload file.

Reviewer 3 Report
The manuscript contains interesting material on problems of quantum mechanics.
Namely there is suggestion to use probability which is related not only to real variables but to extend this notion and application in quantum mechanics formalism to complex plane. The changes of quantum mechanics formalism are always used in connection of known differences of classical mechanics and statistics with quantum ones.
Though the paper is not the final version of constructing the quantum mechanics formalism but authors show that quantum probability given by modulus squared of wave function can be also explained using their formalism of probabilities on complex plane. As initial ideas the paper contains interesting novelty and I recommend it for publication in the Journal. My only comment is that it would be interesting for readers if the approach called probability representation of quantum mechanics see M. Asorey, A. Ibort, G. Marmo, F. Ventriglia, “Quantum Tomography Twenty Years Later”, Phys. ,v. 90, p. 074031 (2015) https://doi.org/10.1088/0031-8949/90/7/074031; V. N. Chernega, S. N. Belolipetskiy, O. V. Man'ko, V. I. Man'ko, "Probability representation of the quantum mechanics and star-product quantization", J. Phys.: Conf. Series, v.1348, 012101 (2019) doi:10.1088/1742-6596/1348/1/012101 ; V. N. Chernega, O. V. Man'ko, V. I. Man'ko, "Probability representation of quantum states as renaissance of hidden variables -God plays coins", J. Rus. Laser Res., v.40, N 2, p.107(2019) doi:10.1007/s10946-019-09778-4 and relation of the probabilities in this approach with probabilities on complex plane will be mentioned in two-three phases. After this minor amendment paper can be published.

Author Response
Reply to reviewer 3 is in the upload file.

Round 2
Reviewer 3 Report
Authors took into account the referee comments. Paper devoted to the quantum mechanical foundations based on using complex variable instead of real variable contains some novelty and is interesting. I suggest publication of the amended paper as it is.